# Regulation of the photophysical dynamics of metal nanoclusters by manipulating single-point defects

Peiyao Pan[1,4], Weinan Dong[2,4], Wentao Huang[1], Xue Bai [2], Zhennan Wu [2] ✉, Xi Kang [1,3] ✉ & Manzhou Zhu [1,3] ✉

Metal nanoclusters have served as an emerging class of programmable nanomaterials with customized structures. However, it remains highly challenging to achieve the single-atom regulation of metal nanoclusters without altering their structural frameworks. Here, we achieve the single-point defects manipulation based upon a cluster pair of $Au_{21}$ and $Au_{22}$ by meticulously complementing the surface defects of the former nanocluster with an additional single-Au complex. The two nanoclusters exhibited identical geometric structures, but their pronounced quantum-confinement effects resulted in different electronic properties, evident in their distinct optical absorption and emission characteristics. Temperature-dependent steady-state photoluminescence spectra and femtosecond transient absorption spectra showed that the manipulation of a single-point defect in $Au_{22}$ inhibited non-radiative decay pathways, reduced electron loss at higher energy levels, and accelerated intersystem crossing, which ultimately enhanced its emission intensity. Overall, the $Au_{21}$ and $Au_{22}$ cluster system in this study provides a cluster platform with controllable surface single-point defects, enabling the regulation of the photophysical dynamics at the atomic level.

Nanoscience has been flourishing since Richard Feynman's groundbreaking speech about the possibilities of manipulating matter at the atomic level, famously titled "There's Plenty of Room at the Bottom"[1,2]. For a long time, it has been the dream of nanoscientists to master atomic-level manipulations and control the structure of nanomaterials precisely. With the ongoing accumulation of synthetic knowledge and the development of advanced analytical methods, researchers can now tailor the composition and morphology of metal nanoparticles[3–6]. However, achieving atomic-level adjustments at specific sites on the nanoparticle surface—such as adding or removing one or two metal atoms at designated positions-remains challenging. These atomic

modifications are crucial, as they control the physicochemical properties of the nanomaterials[7–11].

Metal nanoclusters are an emerging class of promising nanomaterials due to their atomically precise structures[12,13]. Their nanoscale sizes endowed these nanoclusters with molecular-like characteristics, featuring discrete electronic energy levels and strong quantum-confinement effects[14–16]. Indeed, the quantum-confinement effects of metal nanoclusters render them programmable nanomaterials with structure-dependent properties, and any perturbations on compositions/structures may induce variations in clusters' physicochemical performances[5,17–19]. In turn, the atomically precise nature of nanoclus-

---

[1]Key Laboratory of Structure and Functional Regulation of Hybrid Materials of Ministry of Education, Anhui Province Key Laboratory of Chemistry for Inorganic/Organic Hybrid Functionalized Materials, Department of Chemistry, Anhui University, Hefei, China. [2]State Key Laboratory of Integrated Optoelectronics, JLU Region, College of Electronic Science and Engineering, Jilin University, Changchun, China. [3]National Key Laboratory of Opto-Electronic Information Acquisition and Protection Technology, Anhui University, Hefei, Anhui, China. [4]These authors contributed equally: Peiyao Pan, Weinan Dong. ✉e-mail: wuzn@jlu.edu.cn; kangxi_chem@ahu.edu.cn; zmz@ahu.edu.cn

ters enables researchers to master the structure-property correlations, which is essential for visualizing the quantum-confinement effects of these nanomaterials[20–23]. In this context, it is necessary to develop structurally analogous cluster systems with comparable properties for such a visualization, which requires the atomic-level manipulation of nanoclusters[24–30]. However, as of now, it remains highly challenging to achieve the single-atom regulation of metal nanoclusters without altering their structural frameworks[31–35]. Atomic-level understanding of the structure-dependent physical-chemical properties requires newly developed cluster systems as model platforms and precise tools[36–43].

Herein, the atomic-level manipulation has been accomplished using two structurally analogous gold cluster molecules, $[Au_{21}(AmdS)_{12}(PPh_2py)_3]^+$ and $[Au_{22}(AmdS)_{12}(PPh_2py)_4]^{2+}$ (abbreviated to $Au_{21}$ and $Au_{22}$, respectively), with which the photophysical dynamics of metal nanoclusters have been regulated at the atomic level. Specifically, the single-point defect of the $Au_{21}$ nanocluster could be complemented by the addition of a single-gold-atom complex, giving rise to its structural analog, the $Au_{22}$ nanocluster. The maintained structural framework and the single-atom disparity of the two nanoclusters rendered them a platform for visualizing the quantum-confinement effects in determining their photophysical properties. While both cluster analogs maintained a consistent geometric framework, they exhibited evidently different electronic structures and distinct chromatic properties——$Au_{21}$ displayed reddish-brown absorption, whereas $Au_{22}$ showed yellowish-green absorption. Additionally, the $Au_{22}$ showed much brighter photoluminescence (PL) compared to the single-point defective $Au_{21}$, with PL quantum yields of 47.63% for $Au_{22}$ and 13.10% for $Au_{21}$. Such differences in photophysical properties, triggered by the single-atom manipulation, have been unambiguously rationalized using a combination of temperature-dependent steady-state PL spectroscopy and femtosecond transient absorption (fs-TA) spectroscopy. Our findings revealed that the weaker electron-phonon coupling and faster intersystem crossing (ISC) in $Au_{22}$ contributed to its enhanced emission intensity.

## Results

### Synthesis and structural characterization

The $Au_{21}$ nanocluster was synthesized using a one-pot synthetic method by directly reducing Au-AdmS-$PPh_2$py complexes with $NaBH_4$ (Supplementary Fig. 1a; see Methods). The $CH_2Cl_2$ solution of $Au_{21}$ was reddish-brown; however, upon the addition of the $AuPPh_2$pyCl complex to $Au_{21}$, the solution color altered from reddish-brown to yellowish-green, indicating the cluster transformation from $Au_{21}$ to $Au_{22}$, derived from their mass characterizations (Supplementary Fig. 1b,c). To assist the sample ionization in electrospray ionization mass spectrometry (ESI-MS), cesium acetate (CsOAc) was added to the cluster sample. As shown in Supplementary Figs. 1c, 2, two prominent mass signals at $m/z$ of 3549.67 and 3696.46 were detected in the positive mode, which matched well with the chemical formulas of $[Au_{21}(AdmS)_{12}(PPh_2py)_3(CH_3OH)Cs^+]^{2+}$ (calc $m/z$ 3549.68) and $[Au_{22}(AdmS)_{12}(PPh_2py)_4]^{2+}$ (calc $m/z$ 3696.34), respectively. In this context, the $Au_{21}$ and $Au_{22}$ nanocluster molecules were in "+1" and "+2"-charge states, respectively, demonstrating their identical free valence electron numbers of 8e, i.e., 21(Au) − 12(SR) − 1(charge state) = 8 for the $Au_{22}$ nanocluster and 22(Au) − 12(SR) − 2(charge state) = 8 for $Au_{21}$.

Single crystals of $Au_{21}$ and $Au_{22}$ nanoclusters were cultivated by diffusing hexane into their $CH_2Cl_2$ solutions over 7 days, and their atomically precise structures were determined using single-crystal X-ray diffraction (Supplementary Tables 1,2). The $Au_{21}$ cluster crystallized in the orthorhombic space group *Fddd*, while $Au_{22}$ crystallized in a monoclinic system with a *C2/c* space group, resulting in distinct packing arrangements within their respective crystal lattices. Structurally, the two nanoclusters exhibited comparably geometric structures while the surface single-point defect of the $Au_{21}$ nanocluster was complemented by a single-gold-atom complex, giving rise to its

structural analog, the $Au_{22}$ nanocluster (Fig. 1). Specifically, the $Au_{17}$ core of the $Au_{21}$ nanocluster could be conceptualized as consisting of a twisted $Au_{11}$ unit and a twisted $Au_{10}$ unit that share an $Au_4$ face (Fig. 1a). In contrast, the $Au_{18}$ core of the $Au_{22}$ nanocluster was made up of two twisted $Au_{11}$ units by sharing the same $Au_4$ face (Fig. 1b). While the $Au_{10}/Au_{11}$ structures here adopted a cuboctahedral shape, their cuboctahedral configuration was distorted. Additionally, the overall structure of $Au_{21}$, depicted in Fig. 1c,e, featured an $Au_{17}$ kernel protected by four $Au_1(SR)_2$ motifs, four $\mu_2$-SR ligands, and three vertex $PPh_2$py ligands. In comparison, the $Au_{22}$ nanocluster contained the same surface $Au_1(SR)_2$ and $\mu_2$-SR stabilizers as those found in $Au_{21}$, while four $PPh_2$py ligands were anchored at vertex positions of the $Au_{18}$ kernel (Fig. 1d,f). Collectively, the introduced single gold atom would not alter the overall framework of the $Au_{21}$ nanocluster but fill in its surface defects by anchoring an additional $PPh_2$py stabilizer, giving rise to a structurally analogous nanocluster pair with single-point defects manipulation.

Although the introduction of a single gold atom to the surface defect of $Au_{21}$ would not alter its overall skeleton, the corresponding bond lengths underwent readjustment. As depicted in Supplementary Fig. 3a, the average Au-Au bond length of the $Au_{18}$ kernel in $Au_{22}$ was shorter than that of the $Au_{17}$ kernel in $Au_{21}$, measuring 2.899 Å compared to 2.953 Å. Moreover, the single-point defect in $Au_{22}$ caused a notable decrease in both peripheral Au-P and Au-S bond lengths (Supplementary Fig. 3b,c). In this context, at the molecular level, the introduction of a surface single-gold-atom rendered the cluster skeleton more compact, and thus the overall framework of $Au_{22}$ was more rigid. The comparisons of the metal skeleton and the core size of the two clusters also support this perspective. As shown in Supplementary Fig. 4, the length and width of the $Au_{22}$ nanocluster are 9.87 and 10.03 Å, respectively, both shorter than those of the $Au_{21}$ nanocluster. Additionally, the core of the $Au_{22}$ cluster is more compact than that of the $Au_{21}$ cluster. The strengthened structural rigidity of the $Au_{22}$ nanocluster might enhance the emission intensity of nanoclusters in their molecular states by minimizing vibrational relaxation (discussed below)[44,45].

From a supramolecular chemistry perspective, the crystalline packing arrangement of $Au_{21}$ and $Au_{22}$ nanoclusters in the lattice differed significantly. The intercluster distances between adjacent $Au_{21}$ molecules were determined as 20.11 and 22.08 Å, while those for $Au_{22}$ were 17.98 and 19.80 Å, demonstrating that the $Au_{22}$ cluster molecules were more closely packed in the crystal lattice (Supplementary Figs. 5–7). In addition, several intermolecular hydrogen interactions (H•••H) were observed between adjacent $Au_{21}$ nanoclusters with an average interaction length of 2.32 Å (Supplementary Fig. 8a,b). By comparison, the crystal lattice of $Au_{22}$ not only contained several intermolecular H•••H interactions (average interaction length: 2.43 Å) but also included multiple C-H•••$\pi$ interactions (Supplementary Fig. 8c-e). Such rich intermolecular interactions were probably responsible for the more closely packed $Au_{22}$ clusters in the crystal lattice[46,47].

### Photophysical properties

Due to the strong quantum-confinement effect of metal nanoclusters with nanoscale sizes, the single-point defect manipulation would result in differences in the photophysical properties of the $Au_{21}/Au_{22}$ cluster pair. Indeed, the single-atom regulation has been shown to affect the geometric structures of the two nanoclusters, which would ultimately influence their electronic structures and optical performances. As illustrated in Fig. 2a,b, the $CH_2Cl_2$ solution of $Au_{22}$ displayed intense optical absorption around 590 nm, accompanied by a shoulder band at 470 nm, whereas $Au_{21}$ presented only a weak and broad peak at 510 nm. The PL properties of $Au_{21}$ and $Au_{22}$ nanoclusters were then evaluated under ambient conditions. As shown in Fig. 2a,b, strong emissions of $Au_{21}$ and $Au_{22}$ nanoclusters occurred at 715 and 730 nm, respectively, when the cluster solution was excited at 375 nm. The PL

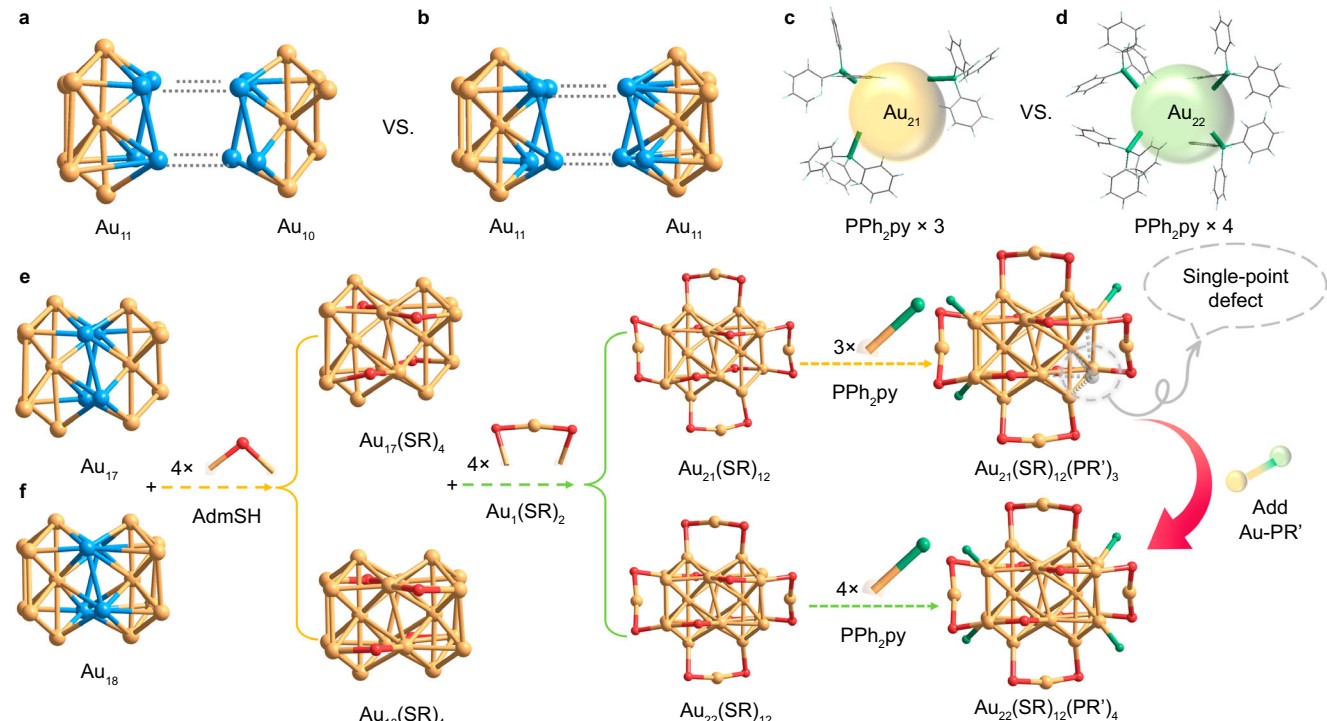

**Fig. 1 | Structural comparison between Au₂₁ and Au₂₂ nanoclusters. a** The Au₁₇ kernel of the Au₂₁ nanocluster comprises one Au₁₁ and one Au₁₀ units by sharing four Au atoms. **b** The Au₁₈ kernel of the Au₂₂ nanocluster consists of two Au₁₁ units by sharing four Au atoms. **c** Three PPh₂py ligands acting as terminals of the Au₁₇ core. **d** Four PPh₂py ligands acting as terminals of the Au₁₈ core. **e** Structural anatomy of the Au₂₁ nanocluster with a peripheral single-atom defect. **f** Structural anatomy of the Au₂₂ nanocluster with a full-protected surface. SR AdmS; PR' PPh₂py. Color legends: light blue sphere and orange sphere, Au; red sphere, S; green sphere, P; light grey sphere, N; grey sphere, C; white sphere, H.

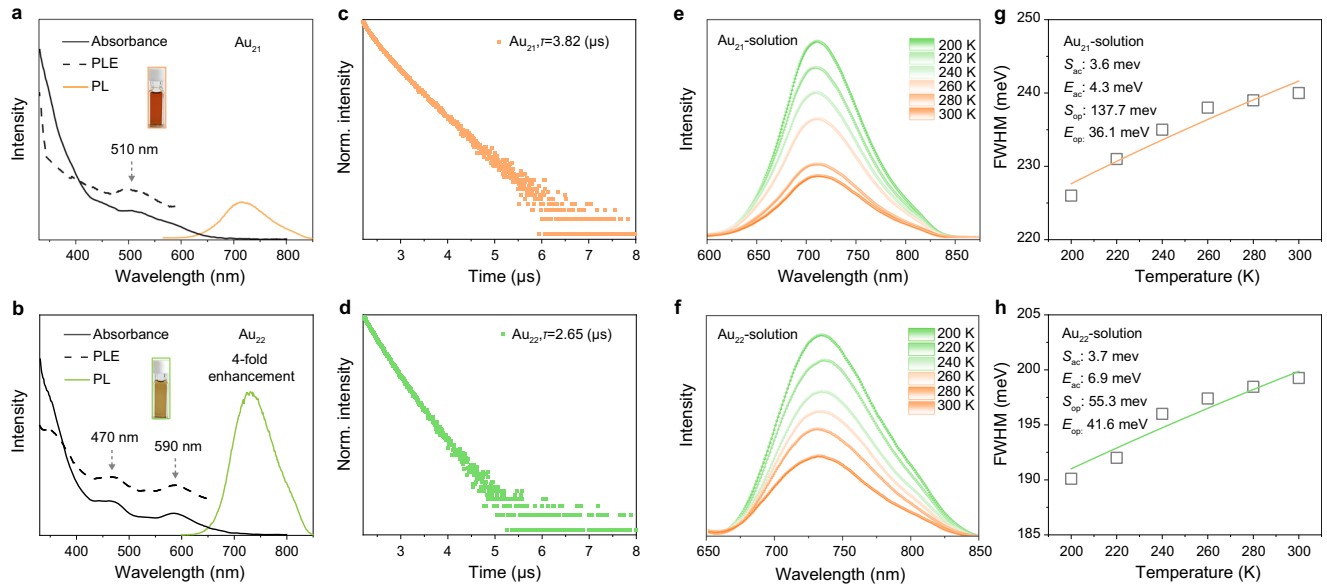

**Fig. 2 | Optical analysis. a** Optical absorptions, excitations (PLE), and emissions (PL) of Au₂₁ nanoclusters. **b** Optical absorptions, excitations (PLE), and emissions (PL) of Au₂₂ nanoclusters. **c** PL lifetimes of Au₂₁ nanoclusters. **d** PL lifetimes of Au₂₂ nanoclusters. **e** Temperature-dependent PL spectra of Au₂₁ nanoclusters dissolved in CH₂Cl₂. **f** Temperature-dependent PL spectra of Au₂₁ and of Au₂₂ nanoclusters dissolved in CH₂Cl₂. **g** FWHM of the steady-state PL spectra as a function of temperature for Au₂₁ nanoclusters dissolved in CH₂Cl₂. **h** FWHM of the steady-state PL spectra as a function of temperature for Au₂₂ nanoclusters dissolved in CH₂Cl₂.

intensity of Au₂₂ was four times greater than that of Au₂₁. In addition, the PL QY of the Au₂₂ nanocluster was determined to be 47.63%, evidently enhanced from the 13.10% of the Au₂₁ nanocluster with a surface single-point defect. The Au₂₂ nanocluster exhibited a refined structure symmetry relative to Au₂₁, which improved the structure rigidity and weakened the framework vibration of the former cluster, resulting in its higher PL intensity. Furthermore, the average PL lifetimes of Au₂₁ and Au₂₂ were measured as 2.65 and 3.82 μs, respectively, and the microsecond lifetimes suggested their analogous phosphorescent characteristic (Fig. 2c,d). To obtain more accurate and direct measurements, we tested the excitation spectra of the two clusters and found a high degree of agreement with their corresponding

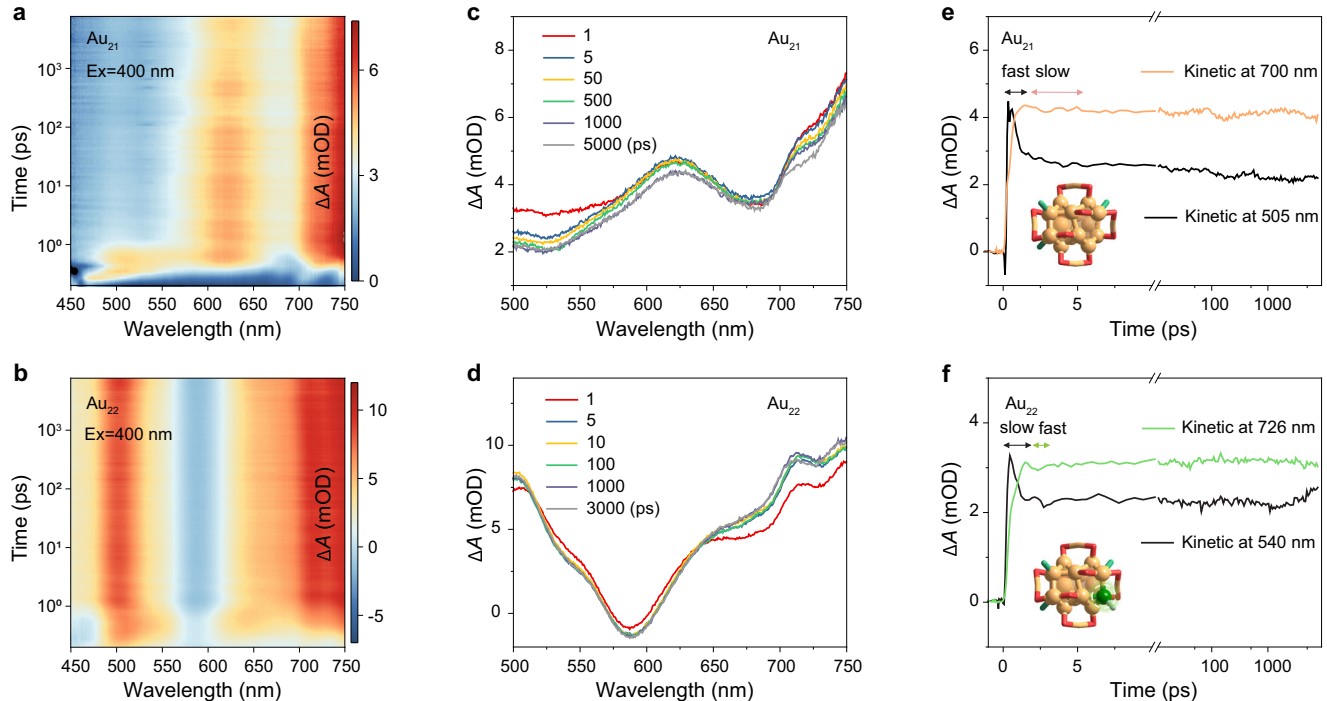

**Fig. 3 | Excited state dynamics of Au$_{21}$ and Au$_{22}$ nanoclusters dissolved in CH$_2$Cl$_2$.** Data map of femtosecond TA of **a** Au$_{21}$ and **b** Au$_{22}$ nanoclusters pumped at 400 nm. TA spectra of **c** Au$_{21}$ and **d** Au$_{22}$ nanoclusters at different time delays. TA kinetic traces selected at specific probe wavelengths of **e** Au$_{21}$ and **f** Au$_{22}$ nanoclusters.

absorption spectra (Fig. 2a,b; see Methods for test conditions). As shown in Supplementary Fig. 9, the excitation-dependent emission spectra indicate single sources of Au$_{21}$ and Au$_{22}$. In this context, the characteristic peaks of the excitation spectra aligned with the absorption spectra of the two nanoclusters, suggesting that the absorption and emission processes occur at the same energy level[48,49].

To investigate the effect of single-point defect manipulation on photophysical and vibration properties of the two gold nanoclusters, we further analyzed their temperature-dependent steady-state PL spectroscopy. A consistent increase in PL intensity was observed for both nanoclusters in their solution state as the temperature decreased from 300 to 200 K (Fig. 2e,f). To better understand the vibrational properties of Au$_{21}$ and Au$_{22}$ nanoclusters, we extracted the temperature-dependent full width at half maxima (FWHM) of their emission peaks, and the plots are given in Fig. 2g,h. The distribution of FWHMs could be well fitted according to Eq. (1)[50–52].

$$\Gamma(T) = \Gamma_0 + \sqrt{S_{ac}E_{ac}\coth\left(\frac{E_{ac}}{2k_BT}\right) + S_{op}E_{op}\frac{1}{e^{\frac{E_{op}}{k_BT}} - 1}} \quad (1)$$

Upon where $\Gamma_0$ was the temperature-independent intrinsic linewidth, Sac and Sop were the coupling strengths for acoustic phonons and optical phonons, respectively, and $E_{ac}$ and $E_{op}$ were the average energy of acoustic phonons and optical phonons. As illustrated in Fig. 2e–h, the core-directed low-frequency acoustic phonons (4.3 meV for Au$_{21}$ and 6.9 meV for Au$_{22}$) barely affected the cluster emission, while the fitted optical phonon energies (36.1 meV for Au$_{21}$ and 41.6 meV for Au$_{22}$) indicated that the Au-S vibrations from cluster surfaces or interfaces dominated their non-radiation. The coupling strength of 55.3 meV for Au$_{22}$ was crucially lower than the value of 137.7 meV for Au$_{21}$, suggesting a weaker electron-phonon coupling and less PL quenching of the former nanocluster.

## Electron dynamics

For molecular-state metal nanoclusters, their PL QYs depend not only on the electron transition of the luminescent state but also on the electron relaxation process in the upper energy levels[53,54]. Femtosecond transient absorption (fs-TA) spectroscopy was then performed to trace the electron trajectory before reaching the luminescent state. Upon the 400 nm excitation and the 500–750 nm detection, two distinct ground-state bleaching (GSB) dents near 520 and 670 nm were obtained in the 2D-TA map for Au$_{21}$ (Fig. 3a,c). Given the total positive signal distribution, the very broad excited-state absorption (ESA) should span the entire probe region, and the differences between absorption and GSB peak positions should arise from the ESA modification. For Au$_{22}$, a main GSB band at 590 nm overlapped with broad ESA was observed, which precisely corresponded to the main absorption peak in steady state, thus reflecting its structure integrity during the measurement (Fig. 3b,d). Of note, the TA signals of Au$_{21}$ and Au$_{22}$ underwent essential changes only for the initial few picoseconds, and then converged to a stable situation.

From the TA kinetic traces selected at specific probe wavelengths of Au$_{21}$ and Au$_{22}$ nanoclusters (Fig. 3e,f), we speculated that at the early part of the TA dynamics, the Au$_{21}$ nanocluster first showed a faster electron injection process and followed by a slower electron decay process compared to Au$_{22}$, finally an electron relaxation process exceeding the detector capacity. Global fitting required three decay components to fit the dynamics (0.36 ps, 2.61 ps, and > 1 ns for Au$_{21}$ and 0.53 ps, 1.11 ps, and > 1 ns for Au$_{22}$) (Supplementary Fig. 10a–d). The <1 ps dynamics could be explained as the internal conversion (IC) of hot electrons from S$_n$ to the S$_1$ state since the values were importantly reduced under the 530 nm pump (Supplementary Fig. 10e, f). Given the phosphorescent characteristic from the triplet state of the two nanoclusters, the few picoseconds were attributed to their intersystem crossing (ISC). The last >1 ns component accounted for the electron−hole recombination because of their µs-level luminescence lifetimes.

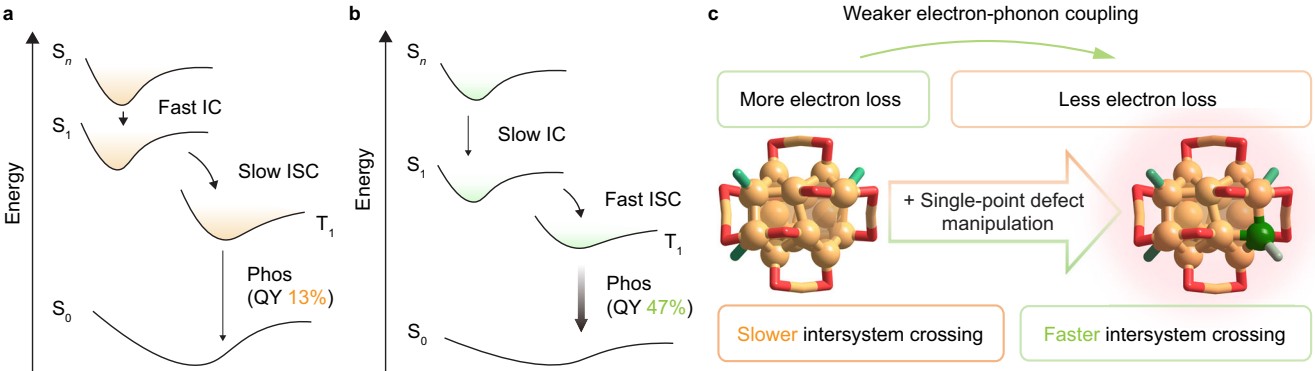

**Fig. 4 | Comparison of PL mechanisms.** PL mechanism of **a** Au$_{21}$ and **b** Au$_{22}$ nanoclusters. **c** Overview of the single-point defect manipulation towards electron-photon coupling and intersystem crossing of Au$_{21}$ and Au$_{22}$ nanoclusters. Color legends: green sphere and orange sphere, Au; red sphere, S; blue-green sphere and light grey sphere, P.

## PL mechanism

In this context, the PL mechanisms of Au$_{21}$ and Au$_{22}$ nanoclusters were proposed. Given the phosphorescence nature of Au$_{21}$ and Au$_{22}$, the excitation light should first pump the ground-state electrons to the excited singlet state, followed by a change in spin direction and eventually relaxed to the luminescent triplet state (Fig. 4a, b). Accordingly, the enhanced PL intensity of the Au$_{22}$ nanocluster relative to Au$_{21}$ could be rationalized from the following two aspects: (i) less energy loss in the upper energy levels, where the slower IC indicated more efficient electron relaxation of Au$_{22}$; 2) faster ISC, which should arise from a reduced energy gap between singlet and triplet for Au$_{22}$ through the single-point defect manipulation rather than the energy-level splitting induced by strong dipole-dipole interaction (Fig. 4c). Collectively, the single-point defect manipulation endowed the structurally comparable Au$_{21}$ and Au$_{22}$ nanoclusters with contrasting photophysical dynamics, and such differences originated from the strong quantum-confinement effects of such small gold nanoclusters.

## Discussion

In summary, the surface single-point defects of the Au$_{21}$ nanocluster could be complemented by an additional single-Au complex, giving rise to its structural analog, the Au$_{22}$ nanocluster with a maintained framework. Although the Au$_{21}$ and Au$_{22}$ nanoclusters exhibited nearly identical geometric structures, their electronic structures were significantly different due to strong quantum-confinement effects. The two nanoclusters manifested distinct photophysical properties, particularly in their optical absorption and emission characteristics. Such differences were rationalized by analyzing their temperature-dependent steady-state PL spectra and femtosecond transient absorption spectra. The single-point defects manipulation on Au$_{22}$ inhibited the non-radiative decay pathways, reduced the electron loss at elevated energy levels, accelerated intersystem crossing, and ultimately enhanced its PL intensity. Collectively, the Au$_{21}$ and Au$_{22}$ cluster system, featuring a controllable single-point defect, provides a platform for visualizing single-atom manipulation effects in determining the photophysical dynamics of metal nanoclusters.

## Methods
### Materials

Adm-SH was prepared following a method reported in ref. 55. All following reagents were purchased from Sigma-Aldrich and used without further purification, including tetrachloroauric(III) acid (HAuCl$_4$·3H$_2$O, 99% metal basis), diphenyl-2-pyridylphosphine (PPh$_2$py), sodium borohydride (NaBH$_4$, 99%), methanol (HPLC grade), ethanol (HPLC grade), dichloromethane (HPLC grade), hexane (HPLC grade), and ethyl ether (HPLC grade).

## Synthesis of [Au$_{21}$(AdmS)$_{12}$(PPh$_2$py)$_3$]$^+$

300 μL of HAuCl$_4$·3H$_2$O (0.2 g mL$^{-1}$) and 50 mg of PPh$_2$py were added into a mixed solvent of 10 mL of CH$_3$OH and 10 mL of CH$_2$Cl$_2$, and the solution was stirred vigorously. After 10 min, a freshly prepared solution of NaBH$_4$ (20 mg in 2 mL of water) was added, and the solution color changed to black immediately. Subsequently, 50 mg of Adm-SH was introduced to the solution. The reaction was proceeded for 8 h, after which the mixture was centrifuged at 10,000 × g for 5 min. The supernatant was collected and evaporated to yield the crude product, which was purified three times with CH$_3$OH. Finally, the precipitate (insoluble in CH$_3$OH) was dissolved in CH$_2$Cl$_2$, giving rise to the solution of the Au$_{21}$ nanocluster, which was used directly in the crystallization process. The yield is 10% based on the Au element (calculated from the HAuCl$_4$·3H$_2$O) for synthesizing the Au$_{21}$ nanocluster.

## Synthesis of [Au$_{22}$(AdmS)$_{12}$(Ph$_2$py)$_4$]$^{2+}$

10 mg of the Au$_{21}$ nanocluster was dissolved in a 20 mL of CH$_2$Cl$_2$, and 1 mg of AuPPh$_2$pyCl complex was added. The solution color changed from red to green within 30 s, indicating the transformation from Au$_{21}$ to Au$_{22}$. The product was purified three times with CH$_3$OH. Finally, the precipitate (insoluble in CH$_3$OH) was dissolved in CH$_2$Cl$_2$, giving rise to the solution of the Au$_{22}$ nanocluster, which was used directly in the crystallization process. The yield is 86% based on the Au element (calculated from the Au$_{21}$ nanocluster) for synthesizing the Au$_{22}$ nanocluster.

## Crystallization of [Au$_{21}$(AdmS)$_{12}$(PPh$_2$py)$_3$]$^+$ and [Au$_{22}$(AdmS)$_{12}$(PPh$_2$py)$_4$]$^{2+}$.

Single crystals of Au$_{21}$ or Au$_{22}$ nanoclusters were cultivated at room temperature by liquid diffusion of *n*-hexane into a CH$_2$Cl$_2$ solution containing the Au$_{21}$ or Au$_{22}$ nanocluster. After 7 days, red block crystals for Au$_{21}$ based on Au and black block crystals for Au$_{22}$ were collected, and the structures of the Au$_{21}$ and Au$_{22}$ nanocluster were determined.

**X-ray crystallography.** The data collection for single-crystal X-ray diffraction (SC-XRD) of all nanocluster crystal samples was carried out on a Stoe Stadivari diffractometer under nitrogen flow using a graphite-monochromatized Cu $K_\alpha$ radiation source ($\lambda = 1.54186$ Å). Data reductions and absorption corrections were performed using the SAINT and SADABS programs, respectively. The structure was solved by direct methods and refined with full-matrix least squares on F$^2$ using the SHELXTL software package. All non-hydrogen atoms were refined anisotropically, and all hydrogen atoms were set in geometrically calculated positions and refined isotropically using a riding model. All crystal structures were treated with PLATON SQUEEZE, and the diffuse electron densities from residual solvent molecules were removed.

## Characterizations

Electrospray ionization mass spectrometry (ESI-MS) measurements were performed on a MicrOTOF-QIII high-resolution mass spectrometer. The sample was directly infused into the chamber at 5 μL min⁻¹. For preparing the ESI samples, nanoclusters were dissolved in $CH_2Cl_2$ (1 mg mL⁻¹) and diluted ($v/v = 1:1$) with $CH_3OH$. All UV–vis optical absorption spectra of the nanoclusters dissolved in $CH_2Cl_2$ were recorded using an Agilent 8453 diode array spectrometer, whose background correction was made using a $CH_2Cl_2$ blank. Nanocluster samples were dissolved in $CH_2Cl_2$ to make dilute solutions, followed by spectral measurement. Photoluminescence (PL) spectra were measured on an FL-4500 spectrofluorometer with the same optical density (OD) of ≈0.1. Absolute PL quantum yields (PL QYs) and emission lifetimes were measured with dilute solutions of nanoclusters on a HORIBA FluoroMax-4P. Femtosecond-TA spectroscopy was performed on a commercial Ti: Sapphire laser system (Spitfire SpectraPhysics; 100 fs, 3.5 mJ, 1 kHz). Solution samples in 1 mm path length cuvettes were excited by the tunable output of the OPA (pump). Excitation-dependent emission spectra involve adjusting the wavelength ($\lambda_{ex}$) of the excitation light and recording the corresponding emission spectrum using an Edinburgh FLS1000 spectrofluorometer. Excitation spectra were conducted on the FL-4500 spectrofluorometer by fixing the emission wavelength and scanning the excitation wavelength.

## Reporting summary

Further information on research design is available in the Nature Portfolio Reporting Summary linked to this article.

## Data availability

The data that support the findings of this study are available from the corresponding authors upon request. Crystallographic data have been deposited at the Cambridge Crystallographic Data Centre (CCDC) under deposition numbers CCDC 2379695 ($Au_{21}$) and 2379692 ($Au_{22}$) and are provided as Supplementary Data 1.

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

## Acknowledgements

We acknowledge the financial support of the NSFC (22371003, 22471001, U24A20480, T2325015 and 12174151), the Ministry of Education, Natural Science Foundation of Anhui Province (2408085Y006), the University Synergy Innovation Program of Anhui Province (GXXT-2020-053), and the Scientific Research Program of Universities in Anhui Province (2022AH030009).

## Author contributions

P.P. and W.H. conceived, carried out experiments, and wrote the paper. W.D. assisted in the synthesis, optical spectral measurement,s and analyzed the data. Z.W., X.B., X.K., and M.Z. analyzed the data and wrote the paper. All authors contributed to the writing of the manuscript.

## Competing interests

The authors declare no competing interests.
