## [Transparent Peer Review file · Nature Communications]

Regulation of the photophysical dynamics of metal nanoclusters by manipulating single-point defects

Corresponding Author: Professor Xi Kang

Version 0:

Reviewer comments:

Reviewer #1

(Remarks to the Author)

In this work, Pan et al. reported the single-point defect manipulation of gold nanoclusters based on a pair of Au₂₁ and Au₂₂ clusters. The structure-optical property correlations have been analyzed, and the photophysical dynamics of metal nanoclusters have been regulated at the atomic level. They concluded that the single-point defect manipulation on Au₂₂ inhibited the non-radiative decay pathways, reduced the electron loss at elevated energy levels, accelerated intersystem crossing, and ultimately enhanced its PL intensity.

Collectively, this work presented an important model for unlocking the realization of high luminescence of metal nanoclusters, and the findings here are interesting. And I believe this work will be of interest to a broad scientific audience, especially to the nanocluster audience. Thus, I would like to suggest the acceptance of this paper in Nature Communications after the authors have addressed the following minor issues.

(A) For metal nanoclusters with small-sized skeletons, such a single-point structural defect would result in large differences in surface units between Au₂₁ and Au₂₂ nanoclusters. The authors provided the comparison of corresponding bond lengths at the molecular level; however, how about the overall frameworks between the two nanoclusters? Is possible that the other units undergo distortion induced by the single-point defect manipulation?

(B) It is accepted that the kernel/surface structure of metal nanoclusters determined their stability largely. How about the stability of the structural comparable Au₂₁ and Au₂₂ nanoclusters? Besides, could Au₂₁ and Au₂₂ nanoclusters maintain their structures after the photophysical measurements, especially the TA test?

(C) Minor suggestion: the single-atom defect in Figure 2 should be changed to single-point defect, because there is an Au-PR difference (not a single-atom).

Reviewer #2

(Remarks to the Author)

In this work, Pan et al. report the synthesis, structure, and optical properties of two gold clusters: [Au₂₁(SR)₁₂(PR₃)₃]⁺ and [Au₂₂(SR)₁₂(PR₃)₄]₂⁺. Interestingly, the two clusters are structurally related but differ by a surface point defect: Au₂₂ has the complete structure, whereas Au₂₁ lacks one surface Au-PR₃ unit. The authors further observed that due to this surface defect, Au₂₁ and Au₂₂ exhibit markedly different absorption and emission properties. Au₂₁ appears red, with a broad absorption band at 510 nm, while Au₂₂ appears green, showing two distinct peaks at 470 and 590 nm. Au₂₂ is also approximately four times more emissive than Au₂₁, with a quantum yield of 47% compared to 13%. This difference in photophysical behavior is further rationalized using temperature-dependent photoluminescence (PL) spectroscopy and femtosecond transient absorption, which indicate that Au₂₂ has weaker electron-phonon coupling and faster intersystem crossing relative to Au₂₁. Overall, this work provides important insights into the effect of surface point defects on the optical properties of gold nanoclusters. However, several issues should be addressed before the manuscript can be considered for publication:

1. Both CIF files appear to correspond to the Au₂₂ cluster with four phosphine ligands. Is there a mistake in the submission files?

2. What are the yields of Au₂₁ and Au₂₂? What is the purity of the as-synthesized Au₂₁ and Au₂₂ clusters? Were any other

cluster species observed prior to crystallization?

3. The ESI spectrum of the Au₂₁ cluster indicates the presence of methanol, yet no methanol is present in the crystal structure. This discrepancy should be clarified. Additionally, what does "AdmSH" represent in Figure S2? Was the simulated formula based on the thiol or the thiolate? Both "AdmSH" and "AdmS" are used; this should be corrected for consistency.

4. The authors mention that the excitation spectra match the absorption spectra (Figure 3D), but it is difficult to verify this from the 3D plots provided. The two panels look very similar. A 2D plot of the excitation spectra should be included for clarity.

Version 1:

Reviewer comments:

Reviewer #1

(Remarks to the Author)

I find the extensive work that has been carried out by this team in addition to the previous submission, very impressive. All issues that have been raised have been adequately addressed, and I very much value that the authors have not felt offended by the criticism, but rather taken this as a chance to optimize the wording and analysis of their experiments. So let me congratulate them for this fine work, and accordingly I strongly recommend to publish this manuscript without any further revisions.

Reviewer #2

(Remarks to the Author)

The authors have answered most of my questions. However, question 4 was not answered properly.

"4. The authors mention that the excitation spectra match the absorption spectra (Figure 3D), but it is difficult to verify this from the 3D plots provided. The two panels look very similar. A 2D plot of the excitation spectra should be included for clarity.

Response: We thank the reviewer for the professional suggestion. We have revised it into 2D plot as suggested. Thank you!"

After examining the new plots, it does not support the authors' claim that "Moreover, the characteristic peaks of the excitation spectra were in coincidence with the absorption spectra of the two nanoclusters, indicating that the absorption and emission processes took place at the same energy level (Figure 3D)." It seems that both Au₂₁ and Au₂₂ plots show two excitation peaks at 375 and 475 nm, which do not match with the absorption spectra of Au₂₁ (with peaks at 510 nm) and Au₂₂ (with peaks at 470 and 590 nm). This needs to be clarified. The 2D excitation spectra at one example emission wavelength (instead of the map) should be provided and plotted together with absorption spectra for better comparison.

Version 2:

Reviewer comments:

Reviewer #2

(Remarks to the Author)

The authors have addressed my questions. I recommend publishing the current version.

We thank all reviewers for their helpful comments. The point-by-point responses are shown in blue in this letter, and revisions are in red. Besides, the revisions in the revised manuscript are highlighted.

Reviewer #1:

In this work, Pan et al. reported the single-point defect manipulation of gold nanoclusters based on a pair of Au₂₁ and Au₂₂ clusters. The structure-optical property correlations have been analyzed, and the photophysical dynamics of metal nanoclusters have been regulated at the atomic level. They concluded that the single-point defect manipulation on Au₂₂ inhibited the non-radiative decay pathways, reduced the electron loss at elevated energy levels, accelerated intersystem crossing, and ultimately enhanced its PL intensity.

Collectively, this work presented an important model for unlocking the realization of high luminescence of metal nanoclusters, and the findings here are interesting. And I believe this work will be of interest to a broad scientific audience, especially to the nanocluster audience. Thus, I would like to suggest the acceptance of this paper in Nature Communications after the authors have addressed the following minor issues.

Response: We thank the reviewer for supporting publication of this work.

(A) For metal nanoclusters with small-sized skeletons, such a single-point structural defect would result in large differences in surface units between Au₂₁ and Au₂₂ nanoclusters. The authors provided the comparison of corresponding bond lengths at the molecular level; however, how about the overall frameworks between the two nanoclusters? Is possible that the other units undergo distortion induced by the single-point defect manipulation?

Response: We thank the reviewer for the insightful suggestion. As suggested, the relevant dissections have been added in the revised Manuscript. Thank you!

We revised the Manuscript into the following statements (Page 6 of the revised Manuscript):

The comparisons of the metal skeleton and the core size of the two clusters also support this perspective. As shown in Supplementary Figure 4, the length and width of the Au₂₂ nanocluster are 9.87 and 10.03 Å, respectively, both shorter than those of the Au₂₁ nanocluster. Additionally, the core of the Au₂₂ cluster is more compact than that of the Au₂₁ cluster.

(B) It is accepted that the kernel/surface structure of metal nanoclusters determined their stability largely. How about the stability of the structural comparable Au₂₁ and Au₂₂ nanoclusters? Besides, could Au₂₁ and Au₂₂ nanoclusters maintain their structures after the photophysical measurements, especially the TA test?

Response: We thank the reviewer for the professional suggestion. By comparing the optical absorptions before and after the transient absorption tests, we found that there were no changes in the characteristic absorption peaks (Figure R1). In this context, the Au₂₁ and Au₂₂ clusters remain stable among the photophysical measurements. Thank you!

Figure R1. Optical absorptions of Au₂₁ and Au₂₂ nanoclusters before and after the transient absorption tests.

(C) Minor suggestion: the single-atom defect in Figure 2 should be changed to single-point defect, because there is an Au-PR difference (not a single-atom).

Response: We thank the reviewer for the helpful suggestion. As suggested, we have changed it to “single-point defect” in Figure 2. Thank you!

Reviewer #2:

In this work, Pan et al. report the synthesis, structure, and optical properties of two gold clusters: [Au₂₁(SR)₁₂(PR₃)₃]⁺ and [Au₂₂(SR)₁₂(PR₃)₄]²⁺. Interestingly, the two clusters are structurally related but differ by a surface point defect: Au₂₂ has the complete structure, whereas Au₂₁ lacks one surface Au-PR₃ unit. The authors further observed that due to this surface defect, Au₂₁ and Au₂₂ exhibit markedly different absorption and emission properties. Au₂₁ appears red, with a broad absorption band at 510 nm, while Au₂₂ appears green, showing two distinct peaks at 470 and 590 nm. Au₂₂ is also approximately four times more emissive than Au₂₁, with a quantum yield of 47% compared to 13%. This difference in photophysical behavior is further rationalized using temperature-dependent photoluminescence (PL) spectroscopy and femtosecond transient absorption, which indicate that Au₂₂ has weaker electron-phonon coupling and faster intersystem crossing relative to Au₂₁. Overall, this work provides important insights into the effect of surface point defects on the optical properties of gold nanoclusters. However, several issues should be addressed before the manuscript can be considered for publication:

Response: We thank the reviewer for supporting publication of this work.

1. Both CIF files appear to correspond to the Au₂₂ cluster with four phosphine ligands. Is there a mistake in the submission files?

Response: We thank the reviewer for the professional suggestion. The uploaded single-crystal files for both nanoclusters are accurate. For the crystal data of Au₂₁, there is an Au-PPh₂py unit following the positional disorder, located in Position 1 or 2 randomly (Figure R2), and the probability of each position is 50%. In this context, there are only three phosphine ligands in the crystal data of Au₂₁, which was further confirmed by ESI-MS measurements. The corresponding discussions were added with the crystal data of the Au₂₁ nanocluster. Please see the newly added discussions with Supplementary Table 1. Thank you!

Figure R2. Positional disorder of Au-PPh₂py unit at Position 1 and Position 2.

2. What are the yields of Au₂₁ and Au₂₂? What is the purity of the as-synthesized Au₂₁ and Au₂₂ clusters? Were any other cluster species observed prior to crystallization?

Response: We appreciate the reviewer for the insightful suggestion. The discussions of synthetic yields and purity of the as-synthesized Au₂₁ and Au₂₂ clusters have been added. Specifically, the synthetic yields of Au₂₁ and Au₂₂ nanoclusters were 10% and 86%, respectively. There might be some by-products for the preparation of Au₂₁, e.g., gold particles and complexes, which were removed by centrifugation and CH₃OH-extraction. No other cluster product (or precipitation) was detected in the crystallization system, demonstrating the purity of the synthesized Au₂₁ nanocluster. For the preparation of Au₂₂, the synthetic yield is very high (86%), suggesting the direct conversion from Au₂₁ to Au₂₂; No other cluster product (or precipitation) was detected except Au₂₂ crystals, demonstrating the purity of the synthesized Au₂₂ nanocluster. Thank you!

We revised the Manuscript into the following statements (Page 11 of the revised Manuscript):

The yield is 10% based on the Au element (calculated from the HAuCl₄·3H₂O) for synthesizing the Au₂₁ nanocluster.

The yield is 86% based on the Au element (calculated from the Au₂₁ nanocluster) for synthesizing the Au₂₂ nanocluster.

3. The ESI spectrum of the Au₂₁ cluster indicates the presence of methanol, yet no methanol is present in the crystal structure. This discrepancy should be clarified. Additionally, what does “AdmSH” represent in Figure S2? Was the simulated formula based on the thiol or the thiolate? Both “AdmSH” and “AdmS” are used; this should be corrected for consistency.

Response: We thank the reviewer for the helpful suggestion. For preparing the ESI samples, nanoclusters were dissolved in CH₂Cl₂ (1 mg/mL) and diluted (v/v = 1:1) with CH₃OH. The CH₃OH in the mass test should come from the solvent. The solvent selection was presented in the Characterization part in the *Supplementary Information*. In the revised version, the corresponding discussions have been added in Supplementary Figure 2. Besides, we have revised the “AdmSH” to “AdmS”. Thank you!

4. The authors mention that the excitation spectra match the absorption spectra (Figure 3D), but it is difficult to verify this from the 3D plots provided. The two panels look very similar. A 2D plot of the excitation spectra should be included for clarity.

Response: We thank the reviewer for the professional suggestion. We have revised it into 2D plot as suggested. Thank you!

Finally, we thank all reviewers for their helpful suggestions and comments.

We thank all reviewers for their helpful comments. The point-by-point responses are shown in blue in this letter, and revisions are in red. Besides, the revisions in the revised manuscript are highlighted.

Reviewer #1:

I find the extensive work that has been carried out by this team in addition to the previous submission, very impressive. All issues that have been raised have been adequately addressed, and I very much value that the authors have not felt offended by the criticism, but rather taken this as a chance to optimize the wording and analysis of their experiments. So let me congratulate them for this fine work, and accordingly I strongly recommend to publish this manuscript without any further revisions.

Response: We thank the reviewer for supporting publication of this work.

Reviewer #2:

The authors have answered most of my questions. However, question 4 was not answered properly. "4. The authors mention that the excitation spectra match the absorption spectra (Figure 3D), but it is difficult to verify this from the 3D plots provided. The two panels look very similar. A 2D plot of the excitation spectra should be included for clarity.

Response: We thank the reviewer for the professional suggestion. We have revised it into 2D plot as suggested. Thank you!"

After examining the new plots, it does not support the authors' claim that "Moreover, the characteristic peaks of the excitation spectra were in coincidence with the absorption spectra of the two nanoclusters, indicating that the absorption and emission processes took place at the same energy level (Figure 3D)." It seems that both Au₂₁ and Au₂₂ plots show two excitation peaks at 375 and 475 nm, which do not match with the absorption spectra of Au₂₁ (with peaks at 510 nm) and Au₂₂ (with peaks at 470 and 590 nm). This needs to be clarified. The 2D excitation spectra at one example emission wavelength (instead of the map) should be provided and plotted together with absorption spectra for better comparison.

Response: We thank the reviewer for the professional suggestion. And, we apologize for the rough result of excitation spectrum, obtained from the cross-section in the excitation-dependent emission spectra (5 nm gap). First, we have included the 2D excitation spectra for one example emission wavelength and compared them with the absorption spectra, where the direct comparison demonstrates a relatively high similarity. To obtain more accurate measurements, we directly tested the excitation spectra of the two emission peaks in Au₂₁ and Au₂₂, finding a high degree of agreement with their corresponding absorption spectra (please see Figure 3D). In this context, the characteristic peaks of the excitation spectra aligned with the absorption spectra of the two nanoclusters, suggesting that the absorption and emission processes occur at the same energy level (Reference: Liu Z. *et al.* J. Am. Chem. Soc. 2023, 145, 19969 and Wang Y. *et al.* J. Am. Chem. Soc. 2023, 145, 26328). Indeed, the difference of directly tested excitation spectra and 2D excitation spectra results from the different excitation conditions, and the former data from the full-spectrum excitation was more accurate. The corresponding discussions have been added in the revised Manuscript and Supplementary Information. Thank you!

We revised the Manuscript into the following statements (Page 7 of the revised Manuscript):

To obtain more accurate and direct measurements, we tested the excitation spectra of the two clusters and found a high degree of agreement with their corresponding absorption spectra (Figure 3D; see Supplementary Information for test conditions). As shown in Supplementary Figure 9, the excitation-dependent emission spectra indicate single sources of Au₂₁ and Au₂₂. In this context, the characteristic peaks of the excitation spectra aligned with the absorption spectra of the two nanoclusters, suggesting that the absorption and emission processes occur at the same energy level.^[49,50]

Please also see the revised Figure 3 and newly added Supplementary Figure 9.

Besides, we added the corresponding test conditions in the Supplementary Information (Page 3 of the revised Supplementary Information):

Excitation-dependent emission spectra involve adjusting the wavelength (λ_{ex}) of the excitation light and recording the corresponding emission spectrum using an Edinburgh FLS1000 spectrofluorometer. Excitation spectra were conducted on the FL-4500 spectrofluorometer by fixing the emission wavelength and scanning the excitation wavelength.

Finally, we thank all reviewers for their helpful suggestions and comments.